# Strength and Performance Tests for Screening Reduced Muscle Mass in Elderly Lebanese Males with Obesity in Community Dwellings

**DOI:** 10.3390/diseases9010023

**Published:** 2021-03-20

**Authors:** Dana Saadeddine, Leila Itani, Andrea P. Rossi, Massimo Pellegrini, Marwan El Ghoch

**Affiliations:** 1Department of Nutrition and Dietetics, Faculty of Health Sciences, Beirut Arab University, P.O. Box 11-5020 Riad El Solh, Beirut 11072809, Lebanon; dana_saadeddine_94@hotmail.com (D.S.); l.itani@bau.edu.lb (L.I.); 2Healthy Aging Center, Department of Medicine, Division of Geriatric, University of Verona, 37126 Verona, Italy; andrea.rossi@hotmail.it; 3Department of Biomedical, Metabolic and Neural Sciences, University of Modena and Reggio Emilia, 41125 Modena, Italy; massimop@unimore.it; 4Clinical Nutrition Unit, Azienda Unità Sanitaria Locale—IRCCS di Reggio Emilia, 42123 Reggio Emilia, Italy

**Keywords:** skeletal muscle mass, muscle strength, obesity, physical fitness, physical performance

## Abstract

The reduction in skeletal muscle mass (SMM) is a common phenomenon in older adults. It is associated with several diseases, a reduction in physical fitness, longer periods of hospitalization and high rates of mortality. We aimed to identify the reliability of simple tools for screening for reduced SMM among older adult males in Lebanon. The Tanita MC-780MA bioimpedance analyzer (BIA) was used to assess body composition in a population of 102 community-dwelling elderly males with overweight or obesity, in order to be then categorized as with or without reduced SMM. Participants also performed the handgrip strength test and the 4 m gait speed test. Of the total sample of 102 participants (mean age 67.4 ± 6.96 years; BMI 30.8 6 ± 4.04 kg/m^2^), 32 (31.4%) met the criteria for reduced SMM. Partial correlation analysis showed that handgrip strength (ρ = 0.308, *p* = 0.002) and 4 m gait speed (ρ = 0.284, *p* = 0.004) were both associated with low SMM. Receiver operating characteristic (ROC) curve analysis identified discriminating cut-off points of 1.1 m/s for the 4 m gait speed test and 32.0 kg for the handgrip strength test. Our study showed that participants displayed a substantial prevalence of reduced SMM. Reduced 4 m gait speed and handgrip strength were associated with low SMM. Clear cut-off points for strength and functional tests for screening for this condition in Lebanese older men were identified.

## 1. Introduction

In the last three decades, much research has been published on body composition in people with overweight and obesity [1,2], particularly focusing on body fat mass and its distribution [3]. However, there is a lack of research on the reduction in skeletal muscle mass (SMM) in this population (i.e., overweight and obesity) [2,4]. In fact, SMM is an important component of a phenotype termed sarcopenic obesity (SO) [5], defined as the increase in body fat mass deposition and the reduction in SMM and muscle strength, which seems to be an ignored outcome and needs further investigation [5].

In the same way, an SMM deficit, especially in older populations, is associated with several serious medical consequences, including low levels of vitamin D [6], osteoporosis and fractures [7], a reduction in physical fitness [8], earlier disability onset, longer periods of hospitalization [9], and mortality [10,11].

Excess weight is a condition defined as an abnormal increased fat deposition and it is associated with an increased risk of chronic morbidity, disability and mortality [12]. The coexistence of overweight and obesity with reduced SMM may exacerbate their respective effects on cardio metabolic disorders [11,13]. A recent study from our group found that SMM reduction in young adult male patients with overweight and obesity was strongly associated with cardiovascular and metabolic diseases [14]. Based on these considerations, the early identification of SMM reduction is of major clinical importance [15]. This will allow the introduction of potential therapeutic strategies, such as dietary and physical activity interventions [16], that aim to limit SMM deterioration in individuals with overweight and obesity, especially among older adults [17].

An important methodological consideration is that the definitions of reduced SMM in overweight and obesity based only on free fat mass (lean mass) value, without considering body mass (body weight) [18], are imprecise. This is a problem, because it is widely known that individuals with overweight and obesity tend to have a relatively higher lean mass [18]. The criteria for low SMM that are based only on the absolute value of free fat mass may not be met in this population, meaning that the prevalence of reduced SMM may be strongly underestimated [18].

The aim of the current study is to assess the prevalence of SMM reduction in Lebanese community-dwelling older adult males with overweight and obesity, using the Oh et al. definition [19], which considers body weight in addition to appendicular SMM [19]. We also aimed to evaluate the reliability of simple measurements, i.e., muscle strength and physical performance tests, in screening for low SMM in this population. Within this scope, the hypothesis is that a high prevalence of low SMM among our population will be found, and this can be easily screened by means of strength and performance tests that do not require specialized instruments.

## 2. Materials and Methods

### 2.1. Participants

The study was conducted in the Department of Nutrition and Dietetics at Beirut Arab University (BAU) in Lebanon, during the period March 2018–February 2020. A total of 102 male participants were recruited from the general population through a simple, random, community e-mail-based survey, sent to members of BAU and other mailing lists. Recruitment focused on self-sufficient older adults of ≥60 years old, living independently in community dwellings. The inclusion criteria were (i) age ≥ 60 years of age, (ii) male and (iii) BMI ≥ 25.0 kg/m^2^. The exclusion criteria were (i) the inability to move without crutches, a walker or other assistive devices, (ii) the presence of artificial limbs or limb prosthesis, and (iii) the presence of active cancer, congestive heart failure, chronic obstructive pulmonary disorder, chronic renal failure, cirrhosis or liver failure.

The study was approved by the Institutional Review Board of BAU (No. 2019H-0063-HS-M-0318), and all participants gave informed, written consent for the use of their anonymized personal data. A questionnaire was administered to elicit information regarding medical history and lifestyle, as well as demographic and social conditions.

### 2.2. Measures

Body weight was measured by a trained dietician involved in the study, using an electronic weighing scale (SECA 2730-ASTRA, Hamburg, Germany). Height was measured using a stadiometer. BMI was then calculated according to the standard formula.

Body composition was measured in the morning in our clinics by a trained dietician, using a segmental body composition analyzer (MC-780MA, Tanita Corp., Tokyo, Japan) [20]. Participants wore their own clothes and weight adjustment for clothing was applied. This method allowed a bioelectrical impedance measurement of the whole body and of each body part (head, trunk, right leg, left leg, right arm and left arm) at a single frequency. Gender, age and height information was then entered into the device and participants were asked to stand in a stable position in bare feet. Their toes and heels were placed in contact with the anterior and posterior electrodes of the weighing platform, respectively. This device provides separate body mass readings for different segments of the body and uses an algorithm incorporating impedance, age and height to estimate the total and regional body fat and fat-free mass. Total fat, lean mass percentages and the appendicular lean mass (ALM) were calculated using standard formulas. Low SMM was defined based on the definition of Oh et al. ((ALM/weight) × 100%) with a cut-off score of 29.60 in males [19].

Participants’ muscle strength and physical performance were evaluated by the handgrip strength test [21] and the 4 m gait speed test [22], respectively, and by adhering to international guidelines. The handgrip strength test was performed as one trial in each hand, using a calibrated dynamometer (Camry Digital Hand Dynamometer Grip Strength, Model EH101–37, China) [23], and the maximum value (kg) was recorded. The 4 m gait speed test was performed on a precisely identified flat course, in a room situated in our clinic, with the 4 m distance marked out with tape [22]. Timing was taken with a chronometer that started when the participant began to move and stopped when he completely crossed the 4 m line with his first foot [22].

Cardiometabolic disease in this study is defined as the presence of any diseases, such as type 2 diabetes, cardiovascular diseases (coronary heart disease, stroke, transient ischaemic attack, and peripheral arterial disease) and dyslipidemia (lowered level of high-density lipoprotein cholesterol, and increased level of low-density lipoprotein cholesterol and triglycerides). These were self-reported to be experienced either simultaneously or separately.

### 2.3. Statistical Analysis

Descriptive statistics are presented as means and standard deviations or frequencies and proportions for continuous and categorical variables, respectively. Data were tested for normality using the Kolmogorov–Smirnov test and the Quantile-Quantile (Q-Q) normality plot. Means were compared via the Student t-test or Mann–Whitney test as appropriate. The chi-squared test for independence was used for proportions. To test the diagnostic performance of muscle strength and gait speed in detecting reduced SMM based on the Oh et al. definition, ALM by weight percentage was used as a gold standard. Classification analysis was done by calculating sensitivity and specificity, and the area under the curve (AUC) of the receiver operating characteristic (ROC) curve. The criterion values of muscle strength and gait speed with maximum sensitivity and specificity were selected for the cut-off points. An AUC > 0.8 indicates an excellent discrimination ability; an AUC of 70–80 is considered acceptable. The cut-off scores achieving 90% sensitivity and their corresponding specificities were also calculated. All values were considered significant at *p* < 0.05.

NCSS 12.0.2 (NCSS, NCSS, LLC. Kaysville, UT, USA) was used for the statistical analysis. Power analysis for the sample size was determined using PASS software (PASS 11. NCSS, LLC. Kaysville, Utah, USA). For the sample of 32 patients classified with reduced SMM vs. 70 with normal SMM, an alpha of 0.05 and an AUC of 0.697 or 0.679, the power was >0.90.

## 3. Results

The sociodemographic and body composition characteristics of the study participants are shown in Table 1. The sample comprised 102 males with a mean age of 67.4 ± 6.96 years, range 60–84 years. All participants were either living with overweight or obesity, with a mean BMI of 30.86 ± 4.04 kg/m^2^. Based on Oh et al.’s definition, 31.4% of the participants were confirmed with reduced SMM. Participants with reduced SMM were more likely to have low income compared to those with normal SMM (75% vs. 51.4%; *p* = 0.025). Those with reduced SMM were significantly heavier (89.10 kg vs. 82.70 kg; *p* = 0.014), with higher BMI (33.20 kg/m^2^ vs. 28.81 kg/m^2^; *p* < 0.0001), body fat (27.45 kg vs. 20.40 kg; *p* < 0.0001) and body fat percentage (30.82 ± 4.40% vs. 24.78 ± 4.13%; *p* < 0.0001), lower gait speed (0.91 ± 0.24 vs. 1.08 ± 0.24 m/s; *p* = 0.002) and lower muscle strength (30.56 ± 6.42 vs. 35.05 ± 6.75 kg; *p* = 0.002).

A significant association was observed for reduced SMM with lower muscle strength (ρ = 0.308, *p* = 0.002) and reduced gait speed (ρ = 0.284, *p* = 0.004) (Figure 1a,b). The results of the ROC analysis for the diagnostic performance are shown in Table 2 and Figure 2a,b. The AUC for muscle strength and gait speed indicates acceptable discriminating ability with an almost 70% chance of discriminating an individual with reduced SMM. The optimal handgrip strength for discriminating reduced ALM was 32.00 kg, with 62.5% sensitivity and 70% specificity, indicating a lower chance for false negatives and false positives. The optimal gait speed cut-off point was 1.01 with a sensitivity and specificity of 63%. The cut-off points at 90% sensitivity were 36.88 kg for handgrip strength and 1.19 m/s for gait speed, with the relatively low specificity of 37.7% and 28.9%, respectively, indicating a higher tendency toward false positives. According to the determined cut-off points for handgrip strength and gait speed, reduced lean body mass (LBM) affected 40.2% and 44% of the study participants, respectively.

## 4. Discussion

In our population of community-dwelling Lebanese older males affected by overweight or obesity, across a wide age range from 60–84 years, the prevalence of low SMM was over 30%. Reduced SMM can be easily screened by means of simple, cheap and accessible measures, i.e., muscle strength and physical performance tests. Individuals with reduced SMM took significantly longer to perform the 4 m gait speed test and demonstrated lower performance in the handgrip strength test than those with normal SMM. We also established cut-off scores for these two measures: a gait speed of 1.1 m/s and a handgrip strength of 32 kg. Individuals with decreased SMM can be easily identified in different settings by means of these simple tests.

The clinical implications of these findings can have a profound impact on health care choices. First, awareness of the prevalence of reduced SMM in older Lebanese adult males with overweight or obesity in community dwellings should be raised through education campaigns using different platforms, e.g., television and social media. Second, the implementation of simple measures in primary care settings, particularly the handgrip strength and 4 m gait speed tests, is extremely important, considering the recent findings on the impact of muscle mass in middle age that can predict the subsequent risk of poor cardiovascular health, especially in males [24]. This is in addition to the association between reduced SMM and the increased risk of insulin resistance [25], as well as the higher levels of circulating inflammatory markers [26]. These functional tests should be considered as initial screening tools to identify those with a high risk of reduced SMM and who would benefit from further investigation via more precise tools, i.e., BIA and dual-energy X-ray absorptiometry (DXA) scans.

Our study has a number of strengths. To our knowledge, it is the first study to assess SMM reduction considering ALM and weight in older adults males with overweight and obesity in Lebanon. Second, our study established cut-off points on both performance tests, which are useful for identifying older individuals who are more likely to have low SMM and who are at higher risk of worsening their health problems. Third, we assessed a reduction in SMM using the Oh et al. definition [19], which has been shown to be of clinical relevance for the Lebanese population [14].

Our study also has some limitations. First, no blood testing was performed; this means that we are not in the position to understand the mechanisms of reduced SMM in this population. Second, our sample is composed only of males and our findings cannot be extended to females of the same population. Third, we assessed body composition using BIA instead of DXA, which is considered the gold standard technique for SMM assessment. However, multi-frequency BIA has been found to be a very accurate measurement and is widely used in different clinical settings. Moreover, it has been shown to be strictly correlated with DXA SMM measurements in healthy adults [27,28], and has been widely validated in individuals with overweight and obesity [29,30]. Finally, the cross-sectional nature of our study has to be considered a further limitation.

## 5. Conclusions

Our study provides evidence that community-dwelling older adult males with overweight or obesity in Lebanon have a substantial prevalence of low SMM. This can be simply screened for by the handgrip strength and 4 m gait speed tests. These tests are easy-to-use measures that can be implemented in several healthcare settings.

## Figures and Tables

**Figure 1 diseases-09-00023-f001:**
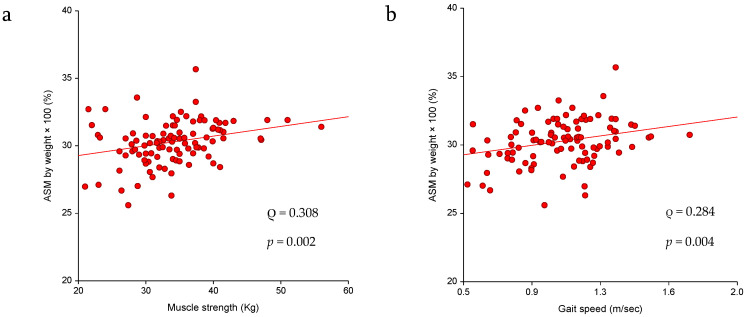
(**a**) Association between low SMM and handgrip strength (kg) and also (**b**) gait speed (m/s) in the study sample (n = 102).

**Figure 2 diseases-09-00023-f002:**
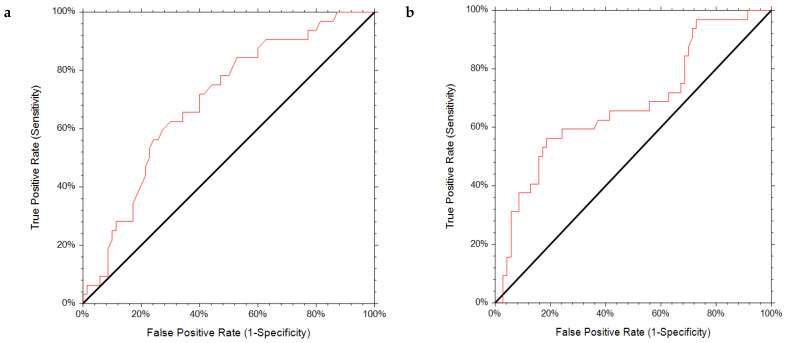
Receiver operating characteristic (ROC) curves for (**a**) handgrip strength test and (**b**) 4 m gait speed test.

**Table 1 diseases-09-00023-t001:** Socio-demographic, anthropometric and cardiometabolic characteristics of the study sample (n = 102).

Demographics	Total N = 102	Normal SMM N = 70	Reduced SMM N = 32	Significance
Age (Years)	67.64 (6.96)	66.13 (5.93)	70.94 (7.96)	*p* = 0.004
Marital status				X^2^ = 3.075; *p* = 0.079
Not married	11 (10.8)	5 (7.1)	6 (18.8)	
Married	91 (89.2)	65 (92.9)	26 (81.3)	
Level of education				X^2^ = 1.155; *p* = 0.283
Lower education	83 (81.4)	55 (78.6)	28 (87.5)	
Higher education	19 (18.6)	15 (21.4)	4 (12.5)	
Employment				X^2^ = 0.073; *p* = 0.787
Not employed	65 (63.7)	44 (62.9)	21 (65.6)	
Employed	37 (36.3)	26 (37.1)	11 (34.4)	
Salary				X^2^ = 5.038; *p* = 0.025
<LL 1 million	60 (58.8)	36 (51.4)	24 (75.0)	
>LL 1 million	42 (41.2)	34 (48.6)	8 (25.0)	
Smoking				X^2^ = 0.415; *p* = 0.520
Non smoker	59 (57.8)	39 (55.7)	20 (62.5)	
Smoker	43 (42.2)	31 (44.3)	12 (37.5)	
Place of residence				X^2^ = 0.164; *p* = 0.686
Outside Beirut	8 (7.8)	6 (8.6)	2 (6.3)	
Inside Beirut	94 (92.2)	64 (91.4)	30 (93.8)	
Weight (kg) ^§^	83.75 (76.78–92.02)	82.70 (75.73–88.25)	89.10 (78.88–106.10)	*p* = 0.014
Body Fat (BF) ^§^	22.25 (18.18–26.42)	20.40 (17.18–27.51)	27.45 (21.78–35.10)	*p* < 0.0001
Body fat percentage (BF%)	26.68 (5.05)	24.78 (4.13)	30.82 (4.40)	*p* < 0.0001
Body mass index (BMI) ^§^	29.89 (28.00–33.07)	28.81 (27.51–31.60)	33.20 (29.58–38.71)	*p* < 0.0001
Gait speed	1.02 (0.25)	1.08 (0.24)	0.91 (0.24)	*p* = 0.002
Handgrip strength	33.64 (6.94)	35.05 (6.75)	30.56 (6.42)	*p* = 0.002
Cardiometabolic disease				X^2^ = 0.012; *p* = 0.913
No	55 (53.9)	38 (54.3)	17 (53.1)	
Yes	47 (46.1)	32 (45.7)	15 (46.9)	

Values are N (%) for categorical variables and Mean (SD) or ^§^ Median (IQR) for continuous variables; X^2^ = Chi Square.

**Table 2 diseases-09-00023-t002:** Diagnostic performance of the handgrip strength (kg) and gait speed cut-off points to detect low SMM in the study population (n = 102).

	AUC	95%CI	*p* Value	Sensitivity	Specificity	Cut Off	Specificity at 90% Sensitivity	Cut-Off at 90% Sensitivity
Muscle strength (kg)	0.696	0.573–0.788	0.0002	0.625	0.700	32.00	0.377	36.88
Gait speed (m/s)	0.679	0.543–0.779	0.0014	0.625	0.629	1.01	0.289	1.19

## Data Availability

Are available from the corresponding author on reasonable request.

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
