# Peer review of "Strength and Performance Tests for Screening Reduced Muscle Mass in Elderly Lebanese Males with Obesity in Community Dwellings"

_diseases, 2021, doi:10.3390/diseases9010023_

Round 1
Reviewer 1 Report
PoczÄ…tek formularza
Journal Diseases (ISSN 2079-9721)
Manuscript ID diseases-1140061
Title: Strength and Performance Tests for Screening Reduced Muscle Mass in Elderly Lebanese Males with Obesity in Community Dwellings
Authors: Dana Saadeddine , Leila Itani , Andrea P Rossi , Massimo Pellegrini , Marwan El Ghoch *
Review:
I read the work submitted to me for review with great pleasure. The research project is very interesting because of the topic problem reduction of muscle mass in the elderly with obesity. Authors showed that it is associated with several diseases, a reduction in physical fitness, longer periods of hospitalization and high rates of mortality. The aim of the study was identify the reliability of simple tools for screening for reduced SMM among older adult males in Lebanon.
I have a few comments which I propose to the authors to improve this work:
Line 35-39: „However, there is a lack of research the reduction in skeletal muscle mass (SMM) in this population [2, 4]. SMM deficit, especially in older populations, is associated with several serious medical consequences including low levels of vitamin D [5], osteoporosis and fractures [6], a reduction in physical fitness [7], earlier disability 38 onset, longer periods of hospitalization [8] and mortality [9, 10].”
Line 43-45: „A recent study from our group found that SMM reduction in young adult male patients with overweight and obesity was strongly associated with cardiovascular and metabolic diseases [13].”
Comment: In the introduction, I propose that the authors introduce a definition of sarcopenia, a disease that is still underestimated and described not only in the elderly, but also secondary sarcopenia described in the course of other diseases (1. Walowski CO, Braun W, Maisch MJ, et al. Reference Values for Skeletal Muscle Mass - Current Concepts and Methodological Considerations. Nutrients. 2020;12(3):755. Published 2020 Mar 12. doi:10.3390/nu12030755, 2. Skrzypczak D, Ratajczak AE, Szymczak-Tomczak A, Dobrowolska A, Eder P, Krela-Kaźmierczak I. A Vicious Cycle of Osteosarcopeniain Inflammatory Bowel Diseases-Aetiology, Clinical Implications and Therapeutic Perspectives. Nutrients. 2021 Jan 20;13(2):293. doi: 10.3390/nu13020293. PMID: 33498571; PMCID: PMC7909530). It is also worth mentioning the concept of sarcopenic obesity (1. Baker JF, Harris T, Rapoport A, et al. Validation of a description of sarcopenic obesity defined as excess adiposity and low lean mass relative to adiposity. J Cachexia Sarcopenia Muscle. 2020;11(6):1580-1589. doi:10.1002/jcsm.12613).
Line 126: The sample comprised 102 males with a mean age of 126 67.4±6.96 years, range 60-84 years. All participants were either living with overweight or obesity, with a mean BMI of 30.86±4.04 Kg/m2
Comment: A very good working assumption, an interesting hypothesis that needs to be developed. The group of obese patients aged over 60 probably has comorbidities, which the authors do not mention. Please prepare and present coexisting diseases grouped in tabular form, especially in terms of CVD and diabetes.
Correlate comorbidities especially with CVD and skeletal muscle mass (SMM) free fat mass (lean mass)
Please describe the medications you are taking.
Line 195: Study also has some limitations.
Comment: A significant limitation of the study is the lack of biochemical data, which the authors reported as a limitation of the study.
According to my opinion this manuscript is suitable for publication after minor revision. The above comments do not detract from the value of the work, it is worth considering them before accepting the final version of manuscript.
Author Response
I read the work submitted to me for review with great pleasure. The research project is very interesting because of the topic problem reduction of muscle mass in the elderly with obesity. Authors showed that it is associated with several diseases, a reduction in physical fitness, longer periods of hospitalization and high rates of mortality. The aim of the study was identify the reliability of simple tools for screening for reduced SMM among older adult males in Lebanon. I have a few comments which I propose to the authors to improve this work:
Line 35-39: „However, there is a lack of research the reduction in skeletal muscle mass (SMM) in this population [2, 4]. SMM deficit, especially in older populations, is associated with several serious medical consequences including low levels of vitamin D [5], osteoporosis and fractures [6], a reduction in physical fitness [7], earlier disability 38 onset, longer periods of hospitalization [8] and mortality [9, 10].”
Line 43-45: „A recent study from our group found that SMM reduction in young adult male patients with overweight and obesity was strongly associated with cardiovascular and metabolic diseases [13].”
Comment: In the introduction, I propose that the authors introduce a definition of sarcopenia, a disease that is still underestimated and described not only in the elderly, but also secondary sarcopenia described in the course of other diseases (1. Walowski CO, Braun W, Maisch MJ, et al. Reference Values for Skeletal Muscle Mass - Current Concepts and Methodological Considerations. Nutrients. 2020;12(3):755. Published 2020 Mar 12. doi:10.3390/nu12030755, 2. Skrzypczak D, Ratajczak AE, Szymczak-Tomczak A, Dobrowolska A, Eder P, Krela-Kaźmierczak I. A Vicious Cycle of Osteosarcopeniain Inflammatory Bowel Diseases-Aetiology, Clinical Implications and Therapeutic Perspectives. Nutrients. 2021 Jan 20;13(2):293. doi: 10.3390/nu13020293. PMID: 33498571; PMCID: PMC7909530). It is also worth mentioning the concept of sarcopenic obesity (1. Baker JF, Harris T, Rapoport A, et al. Validation of a description of sarcopenic obesity defined as excess adiposity and low lean mass relative to adiposity. J Cachexia Sarcopenia Muscle. 2020;11(6):1580-1589. doi:10.1002/jcsm.12613).
Response: Done as suggested. Now we added the definition of Sarcopenic Obesity (Page 1, paragraph 1, introduction section) with a suitable reference (ref.5).
Line 126: The sample comprised 102 males with a mean age of 126 67.4±6.96 years, range 60-84 years. All participants were either living with overweight or obesity, with a mean BMI of 30.86±4.04 Kg/m2
Comment: A very good working assumption, an interesting hypothesis that needs to be developed. The group of obese patients aged over 60 probably has comorbidities, which the authors do not mention. Please prepare and present coexisting diseases grouped in tabular form, especially in terms of CVD and diabetes.
Correlate comorbidities especially with CVD and skeletal muscle mass (SMM) free fat mass (lean mass)
Response: Done as requested, and added the definition of cardiometabolic diseases in the Method section (Page 3, paragraph 4), and their data on cardiometabolic diseases in Table 1.
Please describe the medications you are taking.
Response: The date on medication is not available.
Line 195: Study also has some limitations.
Comment: A significant limitation of the study is the lack of biochemical data, which the authors reported as a limitation of the study.
Response: We thank the reviewer for remarking this important limitation that has been mentioned in the text in the Discussion section (Page 8, paragraph 1).
According to my opinion this manuscript is suitable for publication after minor revision. The above comments do not detract from the value of the work it is worth considering them before accepting the final version of manuscript.
Response: We thank the reviewer for the valuable comments that we did our best to address.
Reviewer 2 Report
Congratulations to the authors for the article, the research is interesting but some improvements should be made.
Two objectives are stated but it would be necessary to start from a study hypothesis.
The material and methods section would be better understood if it were divided into sections such as participants, instruments, procedure...
The discussion does not address much of what is analysed in the discussion in the sense that only 2 articles used in the discussion are cited while 4 articles not used in the introduction are cited.
The depth of the discussion should be improved.
Reference 23 is not dated.
Author Response
Congratulations to the authors for the article, the research is interesting but some improvements should be made.
Response: We thank the reviewer for appreciating our work.
Two objectives are stated but it would be necessary to start from a study hypothesis.
Response: We added the study hypothesis as suggested (Page 2, paragraph 3).
The material and methods section would be better understood if it were divided into sections such as participants, instruments, procedure...
Response: The material and methods section is now divided into sections as suggested.
The discussion does not address much of what is analysed in the discussion in the sense that only 2 articles used in the discussion are cited while 4 articles not used in the introduction are cited.
The depth of the discussion should be improved.
Response: The discussion is now improved as suggested (Page 7, paragraph 2), and we added more references (ref. 24-26).
Reference 23 is not dated.
Response: Now it is added.
Round 2
Reviewer 2 Report
Congratulations to the authors for the improvement of the article.